# Global Analysis of Transcription Start Sites and Enhancers in Endometrial Stromal Cells and Differences Associated with Endometriosis

**DOI:** 10.3390/cells12131736

**Published:** 2023-06-28

**Authors:** Sushma Marla, Sally Mortlock, Sohye Yoon, Joanna Crawford, Stacey Andersen, Michael D. Mueller, Brett McKinnon, Quan Nguyen, Grant W. Montgomery

**Affiliations:** 1The Institute for Molecular Bioscience, The University of Queensland, Brisbane, QLD 4072, Australia; s.marla@imb.uq.edu.au (S.M.); s.mortlock@imb.uq.edu.au (S.M.); b.mckinnon@imb.uq.edu.au (B.M.); quan.nguyen@imb.uq.edu.au (Q.N.); 2The Genome Innovation Hub, The University of Queensland, Brisbane, QLD 4072, Australia; s.yoon@uq.edu.au (S.Y.); j.crawford@imb.uq.edu.au (J.C.); s.andersen2@uq.edu.au (S.A.); 3Department of Gynecology and Gynecological Oncology, Inselspital, Bern University Hospital, University of Bern, 3010 Berne, Switzerland; michael.mueller@insel.ch

**Keywords:** endometrial stromal cells, Cap analysis of gene expression, transcription start sites

## Abstract

Identifying tissue-specific molecular signatures of active regulatory elements is critical to understanding gene regulatory mechanisms. In this study, transcription start sites (TSS) and enhancers were identified using Cap analysis of gene expression (CAGE) across endometrial stromal cell (ESC) samples obtained from women with (*n* = 4) and without endometriosis (*n* = 4). ESC TSSs and enhancers were compared to those reported in other tissue and cell types in FANTOM5 and were integrated with RNA-seq and ATAC-seq data from the same samples for regulatory activity and network analyses. CAGE tag count differences between women with and without endometriosis were statistically tested and tags within close proximity to genetic variants associated with endometriosis risk were identified. Over 90% of tag clusters mapping to promoters were observed in cells and tissues in FANTOM5. However, some potential cell-type-specific promoters and enhancers were also observed. Regions of open chromatin identified using ATAC-seq provided further evidence of the active transcriptional regions identified by CAGE. Despite the small sample number, there was evidence of differences associated with endometriosis at 210 consensus clusters, including *IGFBP5, CALD1* and *OXTR*. ESC TSSs were also located within loci associated with endometriosis risk from genome-wide association studies. This study provides novel evidence of transcriptional differences in endometrial stromal cells associated with endometriosis and provides a valuable cell-type specific resource of active TSSs and enhancers in endometrial stromal cells.

## 1. Introduction

Human endometrium, the inner lining of the uterus, is a vitally important reproductive tissue, essential for fertility and associated with many reproductive disorders [1]. The endometrium consists of distinct cell types: epithelial cells (luminal and glandular) and the supporting mesenchymal cells (stromal cells). The tissue undergoes cyclical changes in cellular composition and gene expression, largely driven by hormonal regulation [2,3,4]. One associated disease is endometriosis, a common gynaecological disease defined by the presence of endometrial glands and stroma outside the uterus. The commonly accepted theory for the origin of cells in lesions is viable endometrial cells being transported in retrograde menstruation [5]. A major approach to discovering the fundamental mechanisms and causes of endometriosis is through identification and understanding of changes in transcriptional regulation associated with disease and the functions of the genetic variants responsible for the genetic component of the disease. Although genome-wide association study (GWAS) results [6] represent a significant breakthrough towards understanding genetic risk factors for endometriosis, the cellular mechanisms through which these variants drive the disease remain largely unknown. 

Understanding gene expression in endometriosis [3] and endometrial cancer, especially in the context of fertility-sparing treatment [7], requires analysis of the transcriptional landscape of several uterine cell types. This approach may help identify disease processes that are unique to certain cell types but would otherwise be averaged out in bulk tissue datasets. One study [8] reported that the development of endometriosis may be influenced by an epigenetic switch that regulates the expression of the GATA isoform in stromal cells. Gene transcription is controlled by functional interactions between promoters and enhancers. Identifying active regulatory regions in the genome is critical for understanding gene regulation and assessing the impact of genetic variation on phenotype. The Cap analysis of gene expression (CAGE) method is a high-throughput, transcriptome-wide and unbiased (towards polyA-tailed RNA) approach, well established at mapping the transcription start sites (TSSs) by capturing the 5′ ends of the transcribed and capped mRNAs [9]. CAGE employs a cap-trapping method and captures 5′ ends of cDNAs, which yields short sequences (tags) of 20 nucleotides that can be mapped back to the reference genome to infer the exact position of the TSSs of captured RNAs. The number of CAGE tags supporting each TSS reflects the relative frequency of its usage and can be used as a measure of expression from that specific TSS, reflecting the activities of a promoter or an enhancer [10]. The functional annotation of the mammalian genome (FANTOM) project [11] generated CAGE expression measurements across 573 primary cell types and tissues. 

To date, there are no publicly available CAGE datasets for endometrium/endometrial stromal cells. In this study, for the first time, CAGE data were produced from endometrial stromal cells of endometriosis patients and controls. We built a genome-wide TSS map for stromal cells and we added a deep ATAC-seq dataset, as well as a total RNA-seq dataset, to the valuable CAGE resource, forming the most comprehensive genomics regulatory database of the endometrial stromal cells. We performed integrative analysis across existing epigenomic datasets to reveal molecular signatures of non-coding stromal cell elements and differences identified in endometriosis patients. 

## 2. Materials and Methods

### 2.1. Samples

Endometriotic biopsies were collected via soft curette (Pipelle de Cornier, Laboratorie CCD, France) from eight women undergoing laparoscopic surgery for suspected endometriosis. Samples were stored in Complete IMDM media (10% foetal calf serum, 1% antibiotics/antimycotics (ThermoFisher Scientific, Waltham, MA, USA)) supplemented with 10% Dimethyl sulfoxide (DMSO) (Thermo Fischer Scientific, Waltham, MA, USA) and stored at −80 °C using the slow-freezing method in a Bicell vessel. The pelvic cavity of each patient was subsequently examined and any suspected endometriotic lesions were removed and disease stage reported according to the revised American Fertility Society staging system (rAFS) [12]. The location of any lesion identified was noted using the ENZIAN classification [13] (Table 1) and subsequently categorized as either superficial peritoneal, ovarian or deep-infiltrating endometriosis (DIE) in the base of the pelvis. 

The endometrial stromal cells (ESCs) were isolated and purified according to our published procedure [14,15]. Cells were cultured/passaged as per standard methods with trypsin/EDTA when approximately 80% confluent. Cells were counted at each passage, population doubling was determined and only cells that remained within a log phase of growth were included in the analysis.

### 2.2. CAGE Sequencing

CAGE data were generated from eight endometriosis stromal cell lines, four of which were derived from women with endometriosis (cases) and four of them from women without endometriosis (controls). For a comprehensive analysis of transcription start sites, we performed two different CAGE protocols, each with its own advantages and disadvantages, as described below.

Commercial protocol: CAGE libraries were generated via the cap-trapping method using a commercially available kit from DNAFORM. The resulting libraries were PCR-free, full-length (un-fragmented) Illumina-compatible libraries containing both polyA and non-polyA transcripts. The entire process from RNA to final library took eight days and required 5 ug RNA/sample with RIN > 7. The kit contained most of the necessary components for the entire workflow.

In-house protocol: CAGE libraries were also generated via the cap-trapping method using an optimized protocol designed and validated in house. This protocol was developed to increase throughput, decrease processing time and offer the flexibility to run samples with standard molecular reagents. The in-house protocol shares many common steps with the commercially available kit from DNAFORM, but it is shorter (taking about five days) and the protocol has a PCR step to generate higher concentration libraries. The resulting libraries were full-length (un-fragmented), Illumina-compatible libraries containing both polyA and non-polyA transcripts that had been minimally amplified by PCR. The entire process required 5 ug RNA/sample with RIN > 7.

Sequencing, read processing and alignment: Sequencing was performed on an Illumina NextSeq500 with 5% PhiX spike-in using the following configurations: Read1-76 bp, Index1-6 bp. Loading concentrations for all the samples were 1.6 pM using a modified denaturation and dilution protocol for low-concentration libraries. Fastq files were generated from the sequencing BCL files using fastqc and raw reads were filtered according to quality using multiqc followed by trimming the reads to remove the ‘N’ base at the 5′ and 3′ ends using MORAI20140528/bin/trimBaseN package. The resulting reads were then aligned to the human genome hg38 assembly https://ftp.ensembl.org/pub/release-100/fasta/homo_sapiens/dna/Homo_sapiens.GRCh38.dna.primary_assembly.fa (accessed on 25 April 2020) using BWA v0.7.15. The uniquely mapped CAGE tags with a minimal mapping quality of 10 were used in this study. The BWA SAM-formatted alignments were converted to BAM format using samtools. The BAM files were then converted to bed files and BigWig files using bedtools2 v 2.24.0 and samtools.

### 2.3. Identification of CAGE-Defined Transcription Start Sites (CTSS) and Tag Clusters (TC)

The vast number of small CAGE tag sequences demands considerable data processing, and statistical approaches are required for biological inferences and discoveries. The follow-up data analysis was based primarily on the annotation of CAGE sequence tags to known transcripts/genes. CAGE data processing, data formats and analysis methodologies are not yet standardised, and the scientific community is still working to develop them. CAGE tags/sequences are aligned to genome sequences using simple computational processes (called BLAST) and then counted, which provides the frequency of RNA expression. CAGE expression is measured specifically for each transcription start site (TSS).

TSSs frequently cluster into tag clusters (TC) that represent transcriptionally active regions [9]. CTSS (CAGE-detected transcription start site) is a cluster of CAGE tags that share the same nucleotide position at their 5′-ends which are defined to reflect a high-resolution map of each individual TSS/TC (tag clusters), spanning a genomic area indicating a group of CTSS. TCs describe a larger area, serving as units for identifying potential core promoters. The shapes of TCs change in a regular manner. In mammalian cells, TCs are often smaller than 10 basepairs (Sharp TSSs) or hundreds of basepairs wide (Broad TSSs) (Figure 1).

The uniquely aligned reads from the bed files were provided as input for the CAGEr pipeline in R [16]. The 5′ end of each read was considered as the single-nucleotide resolution CTSS, and counts were generated for each position to represent the number of unique tags starting from that position. These tag counts were normalised to one million reads (TPM), and CTSS that were within a 20 bp distance were merged to generate TCs. Each TC was considered as an active transcription start site, as a functional unit of a promoter or as an enhancer. Prior to clustering, we filtered out low-fidelity TSSs, i.e., the ones supported by less than 2 normalised tag counts in all of the samples and included only singleton CTSS which had a normalised signal above three. The TCs from all the samples that were within 100 bp of each other were clustered together into a single set of non-overlapping consensus clusters to define the promoter regions.

### 2.4. Correlation of Tag Counts within and between Protocols

We performed Pearson’s correlation for normalised CAGE tag counts per TSS between the samples separately for commercial and in-house data. For robustness, only TSSs with at least five tag counts were considered for calculating the correlation coefficient. The ggcorrplot () function in R was used to plot correlation coefficients between all possible pairs of samples.

We obtained the mean consensus cluster tag counts for each promoter/gene from all the samples in each of the commercial and in-house datasets and performed Pearson’s correlation to calculate correlation coefficients between same-sample pairs obtained from the two protocols.

### 2.5. Merging Data from In-House and Commercial Protocols

To increase TC detection sensitivity, we combined CAGE data from the commercial and in-house protocols. This approach has been widely applied in integrating CAGE datasets. Fastq files from the same sample obtained from two protocols were concatenated. Alignment, mapping, CTSS and TC identification were repeated for the merged files using the method described above.

### 2.6. Promoter Shapes

The CAGEr pipeline was used to evaluate endometrial stromal cell promoters predicted from CAGE clusters. Broad or narrow promoter types were determined by calculating the inter-quantile range of promoter CAGE clusters by assessing the base pair distance between 10% and 90% of a promoter’s total signal. The shapes of the promoter, as determined by the depth of the mapped reads and the width of the clusters, are a unique feature associated with transcriptional regulation activities. 

### 2.7. Enhancer Identification

Enhancers are distal regulatory elements that regulate the transcription of their target genes [17,18]. RNApolII binds to enhancer regions and eRNAs are transcribed bidirectionally from active enhancers [19]. Similar to mRNAs, eRNAs are transcribed by RNApolII and capped in the process, which can be captured by the CAGE method. Enhancers are transcribed in a cell-type-specific manner [20]. Active enhancers can be detected by CAGE signals at the two ends of the enhancers [21]. CAGEfightR [22] was used to analyse the enhancers from our CAGE data. The CTSS coordinates and counts from the CAGEr pipeline were input as bigwig files into the CAGEfightR program. CTSS appearing in only a single sample were removed. The remaining CTSS were normalised to tags per million (TPM) and summed across the samples to yield a pooled CTSS signal. CTSS within 20 bp distance were merged to obtain unidirectional clusters and were required to have 0.3 TPM in at least two samples. Bidirectional clusters were required to have a balance score ≥ 0.95 and to be bidirectional in at least a single sample with at least ≥ 2 counts in at least one sample. Enhancers identified were then annotated to genomic locations (TxDb.Hsapiens.UCSC.hg38.knownGene: Annotation package for TxDb object(s). R package version 4.1.3). Enhancers annotated to introns and intergenic regions were used in subsequent analyses.

### 2.8. Interaction of Transcription Start Sites and Enhancers

TSS-enhancer candidates were identified using the findLinks function from the InteractionSet R package by looking for very closely spaced TSSs and enhancers that had a highly correlated expression within 50 bp distance. This analysis linked promoters and enhancers which were likely to regulate similar genes/pathways.

### 2.9. Differential Promoter and Enhancer Expression

Differential expression analyses were performed using the Bioconductor package DESeq2 [23] in R to build a statistical model and perform contrasts of endometriosis cases versus controls for consensus cluster tag counts and enhancer counts. Fold changes were obtained along with their associated *p*-value. The Benjamini–Hochberg method was used to control the false-discovery rate (FDR) by adjusting *p*-values for multiple testing correction. A promoter or enhancer was defined as significantly differentially expressed if it had a Benjamini–Hochberg-adjusted (FDR) *p*-adjusted < 0.05.

### 2.10. RNA Extraction and Sequencing

The same eight cell lines were subjected to RNA sequencing. The AllprepRNA Mini kit was used to extract total RNA from endometrial stromal cells that were actively growing. Before collecting the mixture in 1.5 mL Eppendorf tubes, Qiagen lysis buffer was added to the cells directly. RNA was then isolated according to the manufacturer′s instructions (Qiagen, Redwood City, CA, USA). RNA was treated with a Turbo DNA-free kit (Thermofisher Scientific, USA) and the RNA concentrations and integrity of each sample were evaluated using the NanoDRopBD-6000 instrument (ThermoFisher Scientific, Waltham, MA, USA). For RNA sequencing, only samples with an RNA integrity number of >8 were used. Stranded RNA-seq libraries were generated using the Illumina Tru-Seq Stranded Total RNA Gold technique, including ribosomal depletion, per the manufacturer′s instructions (Illumina, San Diego, CA, USA). The Illumina HISeq 4000 was used to sequence libraries that were pooled. Low-quality sequencing reads and contaminated HiSeq Illuminata adaptor sequences were trimmed using Trimmomatic v0.36 [24].

### 2.11. Correlation of CAGE-Seq Data with RNA-Seq Data

The average tag counts annotated to the promoter of each gene were correlated to count estimates from RNA-seq data in the same samples to evaluate concordance in gene expression intensities between RNA-seq and CAGE. Pearson’s correlation was used to estimate the correlation between the log-transformed count data of CAGE and RNA-seq. 

### 2.12. ATAC-Seq Using the Omni-ATAC Protocol from Actively Growing Endometrial Stromal Cells

Sample preparation: ATAC-seq was performed on the same eight cell lines. Cells growing in the tissue culture were Trypsinized and the medium was then washed out. The cells were resuspended in cold PBS and counted. Cell pellets of 50,000 cells were resuspended in 50 µL of ice-cold ATAC-Resuspension Buffer (0.1% NP40, 0.1% Tween-20 and 0.01% Digitonin) by pipetting up and down three times. This cell lysis reaction was incubated on ice for 3 min. After lysis, 1 mL of ATAC-seq RSB containing 0.1% Tween-20 (without NP40 or digitonin) was added to wash the cells, and the tubes were inverted three times to mix. Cells were transported to a pre-chilled (4 °C) fixed-angle centrifuge to pellet the nuclei at 600× *g* for 10 min. The supernatant was removed with two pipetting steps (aspirate down to 100 µL with a p1000 pipette and remove final 100 µL with a p200 pipette) and nuclei were resuspended in 50 μL of transposition mix by pipetting up and down six times. The transposition reaction mixture was then incubated at 37 °C for 30 min in a thermomixer with 1000 r.p.m mixing. The transposed DNA was pre-amplified and purified by AMPure DNA magnetic beads. Fragmented DNA was eluted in 20.5 μL RSB containing 10 mM Tris, pH = 7.4. The quality of the libraries was assessed using Bioanalyzer High-Sensitivity DNA Analysis kit (Agilent, Santa Barbara, CA, USA). 

Sequencing, data processing and peak calling: The libraries were pooled and sequenced to a mean depth of 80–100 million reads per sample on NovaSeq 50 bp paired end SP run. The quality of raw reads was checked using FastQC v0.11.7 [25] and MultiQC v1.6 [26]. Low-quality reads and adapter sequences were trimmed from fastQ files using Trimmomatic v0.36 [24]. Reads were mapped to the GRCh38 human reference genome using Bowtie2 v2.3.4.3 [27]. “MarkDuplicates” and “REMOVE_DUPLICATES” functions from Picard (v2.23.4) were then used to remove duplicate reads. Reads were filtered for non-unique alignment (MAPQ > 30) and low-quality mapping (-F 1804) using samtools (v1.1). Using alignmentSieve v3.5, reads were shifted to adjust for the inserted adapter sequences. Finally, peaks were called using MACS2 v2.2.7.1 [28].

### 2.13. Overlap of ESC CAGE Tag Clusters with ESC ATAC-Seq Peaks

CAGE-detected tag clusters were overlapped with ATAC-seq peaks generated in the same samples using BEDtools intersect [29]. Our approach is a stringent method to define regulatory genomic regions, where both CAGE and ATAC-seq signals are present.

### 2.14. Overlap of ESC Consensus Clusters with FANTOM5 and ENSEMBL Databases

Human transcription start site CAGE peaks data generated from 573 tissues and cell lines were obtained from FANTOM5 https://fantom.gsc.riken.jp/5/datafiles/reprocessed/hg38_latest/extra/CAGE_peaks_expression/hg38_liftover+new_CAGE_peaks_phase1and2.bed.gz (accessed on 30 June 2021). ENSEMBL TSS annotations for the human genome (ENSEMBL annotation version hg38) were obtained from ENSEMBL-Biomart. CAGE-detected consensus clusters were overlapped with human TSS from FANTOM5 and ENSEMBL TSS using the intersect BEDtools function -u for unique overlap.

### 2.15. Overlap of ESC Consensus Clusters with Endometriosis GWAS Signals

Endometriosis GWA meta-analysis summary statistics were available from Sapkota et al. 2017. CAGE-detected consensus clusters were overlapped with the SNPs reaching nominal significance (*p* < 1 × 10^−5^) in the GWAS using -u (unique) function in BEDtools. To account for the promoter region, genomic windows of 2 kb were extended on either side of the consensus clusters and overlapped with the GWAS signals using the intersect function in BEDtools. By mapping GWAS signals to promoter and enhancer regions, as well as genes regulated by these regulatory regions, we expected to find potential mechanisms of significant SNPs that were not within the protein-coding regions. 

### 2.16. Pathway Analysis

To gain greater biological insight on differentially expressed consensus clusters and consensus clusters that were not annotated to FANTOM5 data, we used clusterProfiler [30], which has the ability to analyse and visualize data for enrichment analysis. Gene lists included those identified from the differential consensus cluster expression analysis and those annotated to the potential ESC specific consensus clusters not present in FANTOM5. Benjamini–Hochberg-corrected *p*-values of <0.05 were considered significant enrichment.

## 3. Results

### 3.1. Comparing Quality of the Commercial and In-House CAGE Sequencing Protocols 

Raw tag counts were normalised to quantify the expression from each individual TSS and to enable comparison between multiple samples. Tag counts ranged from 10–20 million for the commercial protocol and 5–9 million for the in-house protocol (Table 2). The CAGEr package was used to count the number of CAGE-detected transcription start sites (CTSS normalised to the sequencing depth) in each sample processed using the in-house and commercial protocols. It was observed that the number of CTSS counts obtained from in-house protocol samples (mean = 95,000) were more than the commercial protocol samples (mean = 75,000) (Figure 2a). The relationship between CAGE tag counts in different samples of each protocol was assessed using Pearson’s correlation to calculate correlation coefficients (r^2^) between all pairs of samples. Samples processed using the commercial protocol were highly correlated, r^2^ ≥ 0.8, as were those processed using the in-house analysis, r^2^ ≥ 0.8. 

### 3.2. Genomic Distribution of CAGE-Defined Transcription Start Sites (CTSS)

To investigate the genomic distribution of CAGE tags, CTSS were annotated to the human hg38 reference genome https://ftp.ensembl.org/pub/release-100/fasta/homo_sapiens/dna/Homo_sapiens.GRCh38.dna.primary_assembly.fa (accessed on 25 April 2020). Distribution of CTSS counts across all the samples and between the two protocols were found to have similar genomic distribution (Figure 2b). As anticipated, the majority (>80%) of CTSS were annotated to promoter regions, followed by an average of 10% annotated to exons, 5% annotated to introns and 2% intergenic.

### 3.3. Identification of ESC Promoter Elements

TSSs in close proximity give rise to functionally equivalent sets of transcripts and are the elements of the same promoter/enhancer. As such, to understand the transcriptional activity of a promoter, CTSS were clustered into tag clusters (TCs). An average of 8313 tag clusters were identified using the commercial protocol and 7770 tag clusters from the in-house protocol. To compare genome-wide transcriptional activity across the samples and to perform expression profiling, tag clusters were aggregated from all the samples into a single set of non-overlapping consensus clusters. Consensus clusters observed from in-house and commercial protocols were 6588 and 7136, respectively. These consensus clusters were annotated to genes. The relationship between consensus cluster tag counts obtained from different samples in each protocol was assessed using Pearson’s correlation to calculate correlation coefficients between all pairs of samples. Consensus cluster tag counts from samples processed using the commercial protocol were highly correlated (r^2^ ≥ 0.8) with counts from samples processed using in-house protocol. 

Despite the use of two protocols in generating the CAGE data from ESCs, results of downstream analyses were consistent between both the protocols. Considering the cost and time, our inhouse protocol generated equivalent results to the commercial protocol but in a shorter time and for less cost. 

### 3.4. Concordance between Sequencing Technologies

To determine if expression of TSSs, as estimated by the CAGE data, is consistent with overall gene expression in ESCs, tag counts were correlated for each gene for the single consensus clusters (annotated to genes) with RNA-seq data generated from the same eight ESC lines. It was observed that consensus cluster counts were significantly (*p* = 2.2 × 10^−16^) positively correlated (cor = 0.334) with RNA-seq data annotated to the same genes.

### 3.5. Transcriptional and Regulatory Elements Are Supported by ATAC-Seq Data

Tag clusters and enhancers identified from CAGE were overlapped with the chromatin accessibility data profiled using ATAC-seq. It was observed that almost 90% of the tag clusters and enhancers fall in open chromatin regions (Table 3). 

### 3.6. Differential Expression of Promoters between Endometriosis Cases and Controls

Given the consistency between the two CAGE protocols, and to maximise read depth for further analysis, the data from the two methods were combined and re-analysed. An average of 8351 tag clusters (Table 4) and 7125 consensus clusters were generated from the merged data and the genomic distribution of CTSS counts was consistent with that observed for the commercial and in-house data separately. As expected, we observed two distinct populations of promoters, defined as sharp (interquartile range < 10 bp) and broad promoters (interquartile range > 10 bp) (Figure 3).

Differential expression analysis resulted in 210 consensus clusters which were significantly differentially expressed between endometriosis cases and controls following correction for multiple testing (FDR-adjusted *p*-value < 0.05) (Table 5 and Figure 4a). After Bonferroni correction for multiple testing (*p* < 7.01 × 10^−6^), 53 out of 210 consensus clusters remained significant (Table 5 and Figure 4a). Eight consensus clusters annotated to the gene insulin-like growth factor binding protein-5 (*IGFBP5*) and two annotated to Caldesmon 1 (*CALD1*) had significantly lower expression in cases compared to controls. Consensus clusters associated with genes for oxytocin receptor (*OXTR*) (Figure 4c) and pregnancy-specific beta-1-glycoprotein 4 (*PSG4*) also had lower expression in endometriosis cases in comparison with controls. 

To gain further understanding into the biological mechanisms underlying the consensus clusters, pathway analysis was performed using the 210 significantly differentially expressed consensus clusters using gene ontology databases (Figure 4b). This revealed significantly enriched processes involved in focal adhesion (*p* = 3.8 × 10^−7^) and cell substrate adhesion (*p* = 4.22 × 10^−7^). 

Annotations of consensus clusters to the genes *IGFBP5, CALD1* and *OXTR*, and their overlap with open chromatin and ensemble regulatory regions, are shown in Figure 5. The majority of clusters annotated to *IGFBP5* were located in the 3′ and 5′ UTR regions overlapping regions of open chromatin and predicted promoter and enhancer regions (Figure 5a). *IGFBP5* clusters differentially expressed between cases and controls were located in promoters and enhancers. Clusters annotated to the *CALD1* gene were observed in the 5′UTR, promoter and open chromatin regions near exonic regions. The differentially expressed clusters between cases and controls were found to overlap with enhancers and promoters. Only one cluster was discovered to be differently expressed between cases and controls in the 5′ UTR region of the *OXTR* gene, which overlapped with the promoter region.

### 3.7. Predicted Enhancers and Differential Expression Analysis between Endometriosis Cases and Controls

Using the CAGEfightR package, 1561 enhancers were identified. Of the 1561 predicted enhancers, 1081 and 480 enhancers were annotated to intron and intergenic regions, respectively. Following interaction analyses, 1176 predicted enhancer–TSS interactions were found where expression of a TSS and nearby enhancer was significantly correlated; however, none of the 1176 interactions remained significant following correction for multiple testing (FDR < 0.05). 

Differential expression analysis resulted in two enhancers which were significantly differentially expressed between endometriosis cases and controls following correction for multiple testing (FDR adjusted *p*-value < 0.05) located in the intergenic region nearby to the genes *LINC02547* and *HTR1D*. 

### 3.8. ESC-Specific Promoters and Enhancers

To determine if consensus clusters identified in ESC were consistent with previously reported TSSs and enhancers, consensus clusters identified in this study were compared with human TSS data from FANTOM5. The FANTOM5 dataset consisted of 1,048,125 TSSs from 573 tissues and cell types. It was observed that 96.5% (6880) of the CAGE consensus clusters overlapped with the FANTOM5 TSSs. The remaining 244 consensus clusters that did not overlap with the FANTOM5 data were annotated to 194 genes, some of which were found to have multiple promoters; for example, there were eight novel promoters annotated to *COL4A2*, six to *MALAT1*, five to *MYH9* and two to *IGFBP5* genes. Of the 244 consensus clusters, 31 were found to be annotated to unknown promoters. Pathway analysis for the 194 genes revealed significantly enriched processes involved in focal adhesion (*p* = 1.34 × 10^−27^) and cell substrate adhesion (*p* = 1.34 × 10^−27^).

Enhancers identified in the ESC were compared with previously reported enhancers from the FANTOM5 to determine if results were consistent. The FANTOM5 dataset consisted of 63,285 enhancers. It was found that 60% of the ESC enhancers overlapped with FANTOM5 enhancers.

### 3.9. Comparison of ESC CAGE Data with ENSEMBL TSS Annotation

CAGE-detected consensus clusters were evaluated against ENSEMBL TSS annotations for the human genome (ENSEMBL annotation version hg38). Out of 7125 consensus clusters that were within the regions, only 2948 matched perfectly with the ENSEMBL TSS annotations. After extending the regions by 100 bp either side, an average of 4293 tags overlapped.

### 3.10. Evidence of Transcriptional Elements in Endometriosis Risk Regions

Consensus clusters and enhancers from endometrial stromal cells were overlapped with endometriosis risk SNPs. None of the consensus clusters or enhancers mapped to known GWAS signals. By extending the genomic window around the consensus clusters by 2 kb to account for the promoter region, consensus clusters were found to overlap 11 endometriosis-associated SNPs across 6 genomic regions (Table 6), including gene regions near *FGD6*, *ARL14EP* and *CDC42* (Figure 6).

## 4. Discussion

Gene transcription is controlled by functional interactions between promoters and enhancers [31]. Identifying tissue-specific molecular signatures of active regulatory elements is critical to understanding gene regulatory mechanisms. A comprehensive understanding of factors affecting the regulation of transcription in the endometrium and endometrial cell types is important for the accurate analysis and interpretation of differential gene expression data from endometrium across biological and disease contexts because the endometrium is the likely source of cells initiating endometriotic lesions. In this study, for the first time, CAGE data were generated from endometrial stromal cells of endometriosis patients and controls to map genomic regulatory regions, including promoters and enhancers in endometrial stromal cells and the location of endometriosis risk loci within genomic regulatory regions. 

Profiling TSSs using CAGE in eight endometrial stromal cells identified a consensus set of ~7000 reproducible TCs (or promoters) and 1500 enhancers, consistent with numbers of previously identified TCs from human pancreatic islets [32]. ESC TCs were enriched to occur in stromal cell promoter states and ATAC-seq peaks, which reflects the expected chromatin landscape at regions where transcription initiation occurs. Theoretically, it is understood that gene expression data obtained from different technologies should be consistent if they measure the same outcomes. CAGE-profiled gene expression measured in this study was comparable with the RNA-seq data from the same samples, as evidenced by a significant positive correlation. However, as expected, the genes with low expression were poorly quantified with CAGE profiling, and this probably reflects the modest sequencing depth in these samples [33]. Using CAGE, the FANTOM consortium has mapped TCs across multiple tissue and cell types from mice and humans [11]. Comparison of endometrial stromal cell TCs with those identified from FANTOM5 data showed a high degree of concordance and revealed that 6.3% of TCs were potentially stromal-cell-specific.

Despite the small sample size, there was evidence of endometriosis case–control differences at 210 consensus clusters. Of the 210 consensus clusters showing differential expression, 53 promoters passed Bonferroni correction, including genes (*IGFBP5*, *OXTR CALD1*) previously associated with endometriosis [15,34,35]. Of particular interest is the differential expression of several TCs for *IGFBP5*. *IGFBP5* is one of a family of binding proteins involved in the regulation of insulin-like growth factor (IGF) signalling. Of the 15 consensus TCs observed for this gene, 13 TCs showed FDR-significant differential expression between endometriosis cases and controls. It was observed that 5 out of 284 TCs which were not annotated to FANTOM5 data belonged to *IGFBP5*. *IGFBP5* showed consistently low expression in endometrial stromal cells obtained from endometriosis patients. *IGFBP5* is selectively expressed in the proliferative phase of human endometrium, indicating that *IGFBP5* may play a role in promoting endometrial cell proliferation [36]. Previous studies suggest that *IGFBP5* is influenced by oestrogen and is consistently overexpressed in endometriotic tissue [37]. Differential expression of *IGFBP5* in bulk endometrial tissue was not observed [3,4]. However, a recent study [38] found that *IGFBP5* was significantly downregulated in a sub-population of stromal fibroblasts that was associated with endometriosis cases. Discrepancies in the level of expression of *IGFBP5* between eutopic and ectopic endometrial tissue and in endometrial cell types is consistent with reports that *IGFBP5* is expressed in a wide range of cell types but exhibits a broad range of biological functions depending on the context [38]. Studies have shown that *IGFBP5* can both positively and negatively regulate IGF signalling in different cell types and tissues and similarly it has been shown to both inhibit and promote cell survival, proliferation and migration [38]. The potential for dysregulation of *IGFBP5* to influence cell survival and proliferation highlight this gene as a good candidate for endometriosis.

Twelve consensus clusters were identified for *CALD1* in endometrial stromal cells, six consensus clusters showed significantly low expression in women with endometriosis compared to women without endometriosis. The *CALD1* gene encodes the protein caldesmon found in the cytoskeleton and plays an important role in smooth muscle contraction and relaxation, as well as cellular functions, such as proliferation, adhesion and cell motility [39,40]. *CALD1* isoforms have also been associated with tumour malignancy in several cancers [41,42,43,44]. Previous studies have shown increased expression of the *CALD1* gene in endometriotic tissue obtained from women with endometriosis; however, they also report a significantly lower expression of the caldesmon protein in eutopic endometrium of women with endometriosis, which is consistent with the lower levels of expression observed in this study [40,45,46]. It has been hypothesised that low levels of caldesmon protein in eutopic endometrium may increase the motility and invasiveness of the endometrium via the inability of the protein to bind to actin and inhibit the activity of actomyosin ATPase [40]. Alternatively, caldesmon activity is also negatively regulated by calmodulin (*CALM2*), which has increased expression in ectopic lesions alongside *CALD1*. Increased expression of *CALM2* may also prevent caldesmon from binding to actin, allowing lesions to have a greater invasive potential [40]. The potential role of *CALD1* in endometriosis pathogenesis in both the context of eutopic and ectopic endometrial cell types warrants further functional investigation.

This study also supports an association between the variation in expression of *OXTR* and endometriosis. One of the consensus clusters for *OXTR* was significantly different between women with and without endometriosis. Oxytocin and its receptor appear to play an important role in the regulation of contractions in the uterus [47]. *OXTR* is expressed in the smooth muscle cells and epithelial cells in the uterus and endometriotic lesions [48]. We show that *OXTR* is also expressed in stromal cells. Studies have shown variable expression of *OXTR* across the menstrual cycle [3,35,49] and in response to endometrial pathologies, including endometriosis, adenomyosis and recurrent implantation failure [35,50,51].

CAGE identified 1561 elements showing enhancer activity located in intron and intergenic regions. Transcribed enhancers were identified across a wide variety of human cells and tissues from the analysis of the FANTOM5 dataset [21]. In this study, 60% of the identified enhancers overlapped with the FANTOM5-identified enhancers. Potential tissue/cell-type-specific enhancers identified in this study may provide valuable insights into regulatory elements that play a distinct role in endometrial structure, function and disease. Ninety-three percent of the CAGE-defined enhancers were enriched in the open chromatin regions, providing further evidence for the presence of potential cell-type-specific enhancers.

Endometriosis is driven by both genetic and environmental factors. A significant breakthrough in genetic research on endometriosis came through GWAS, where 27 genomic risk regions have been identified [6]. However, many implicated variants are classified as non-coding and their significance can be realized only when the associated DNA sequence variants in the tissues relevant to endometriosis are determined. There was no evidence that risk variants associated with endometriosis risk were located within tag clusters or enhancers in endometrial stromal cells; however, there is evidence that functional sites regulating the activity of promoters are concentrated within a few thousand bases of the TSS [52]. Analysis of SNPs with regulatory effects on the expression of neighbouring genes shows that most variants with a significant effect are located within 2 kb of a TSS [52]. Considering the length of these regulatory regions, 11 GWAS SNPs were found to overlap within the 2 kb genomic window of the identified consensus clusters (Table 5). A few of the important gene regions identified here were near *FGD6*, *ARL14EP* and *CDC42*. In relation to endometriosis, several studies have reported association of endometriosis risk with increased expression of *VEZT* and *FGD6* [3,4]. Both *VEZT* and *FGD6* play an important role in plasma membrane, cell adhesion and cytoskeletal remodelling, which are important in lesion formation [53,54]. A recent study suggests the evidence of bidirectional promoters for *VEZT* and *FGD6* [4]. The RNA-seq data revealed that both *FGD6* and *VEZT* were expressed in endometrial stromal cells, although *VEZT* was expressed at a higher level than *FGD6*. From the CAGE data, consensus clusters were only detected for *VEZT* in this investigation. The absence of *FGD6* expression may be attributed to insufficient sequencing depth.

Active TSSs in ESCs in the *FSHB* endometriosis risk locus were also identified. The region upstream of the promoter of *FSHB* (follicle-stimulating hormone subunit B), regulating *FSH* (follicle-stimulating hormone) concentrations, has been associated with risk of endometriosis and several other gynaecological traits and diseases [55]. *ARL14EP* is located on the same chromosomal locus as *FSHB*. *ARL14EP* is expressed in many tissue types, with relatively high levels in the ovary, testis and uterus, and plays a role in controlling the export of major histocompatibility class II molecules along the actin cytoskeleton [56,57]. *FSHB* is only expressed in the pituitary gland and is the beta subunit for FSH, a gonadotropin with key regulatory roles in reproductive function [58]. 

Endometriosis risk alleles in the chr1 region have been associated with increased expression of *CDC42* [59], and active TSSs in ESCs were also located in the *CDC42* endometriosis risk locus. *CDC42* is a member of the Rho family of small GTPases and is thought to regulate a variety of cellular processes, including cell cycle progression, cell polarity, cytoskeletal reorganization and transcription [60]. GnRH-activated *CDC42* regulates FSH and LH in response to pulsatile GnRH [61], there is also evidence that *CDC42* regulates ovarian reserve, follicle activation and granulosa cell function [62], suggesting that altered CDC42 regulation may be implicated in fertility issues associated with endometriosis. 

This study has a number of strengths, the most notable being the generation of the first CAGE dataset generated from ESC, providing cell-type specific data on gene regulatory mechanisms. The ability to compare this dataset with other OMIC datasets generated from the same samples (ATAC-seq, RNA-seq, genotypes) and other publicly available datasets (e.g., FANTOM5) provides additional evidence for the validation of gene regulatory mechanisms and facilitates the identification of 244 TSSs that are not observed in the FANTOM5 datasets. These TSSs may be specific to endometrial stromal cells and has led to the identification of some active TSSs with distinct transcriptional regulation patterns linked to endometriosis status. Differential regulation of these genes was not previously detected in bulk endometrial tissue. Further studies of differential regulation of IGFBP5, CALD1 and OXTR in specific endometrial cell types with disease status will be required to validate the reported results. The power of this dataset is, however, limited by the small sample size and limited sequencing depth. Individual results have not been validated and will need confirmation from alternative methods such as RT-PCR, Western blotting or immunohistochemistry. 

Characterising active TSSs in endometrial stromal cells revealed differences in transcriptional regulation associated with endometriosis that have not been observed in bulk endometrial tissue. This study highlights the importance of mapping cell-type-specific transcriptional events that may contribute to disease processes and provides a unique resource to investigate transcriptional regulation in a cell type important in many reproductive traits and conditions.

## Figures and Tables

**Figure 1 cells-12-01736-f001:**
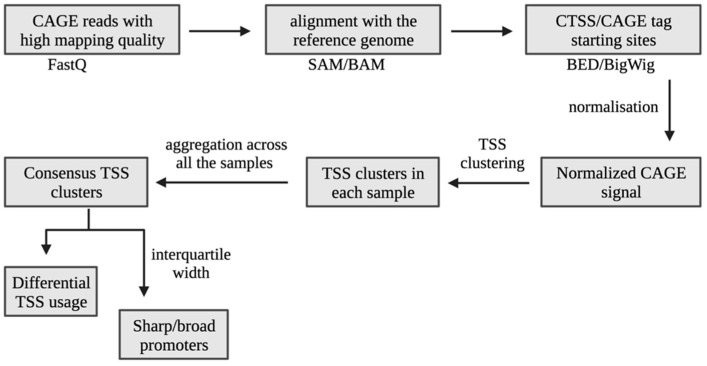
An overview of the CAGE bioinformatics analysis workflow.

**Figure 2 cells-12-01736-f002:**
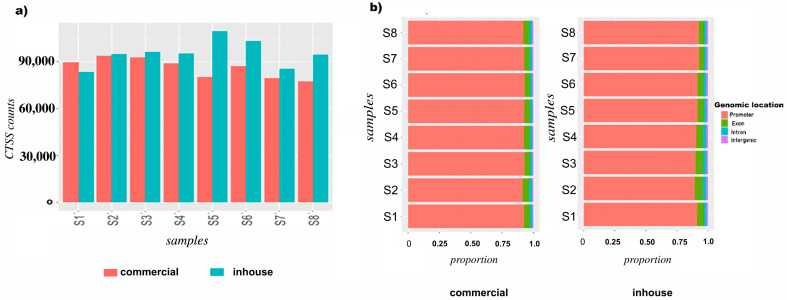
CAGE-identified transcription start sites (CTSS) and genomic distribution of CAGE tags of endometrial stromal cells obtained from commercial and in-house protocols. (**a**) CTSS count comparisons between commercial (orange) and in-house (blue) protocols of CAGE data. X-axis represents sample numbers of endometrial stromal cell lines and Y-axis represents number of CTSS counts; (**b**) genomic distribution of CAGE tags of endometrial stromal cells obtained from commercial and in-house protocols.

**Figure 3 cells-12-01736-f003:**
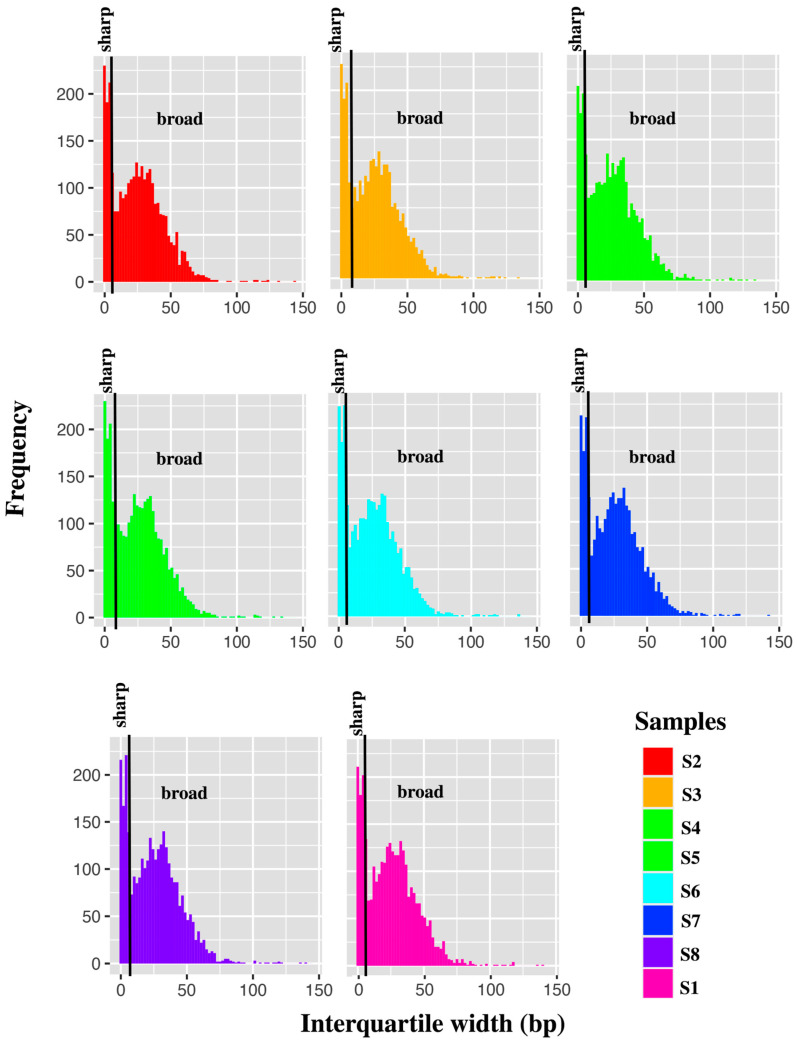
Endometrial stromal cells show two types of promoters: those that initiate transcription over a narrow region (<10 bp, sharp promoter) or over a broad range (10–100 bp).

**Figure 4 cells-12-01736-f004:**
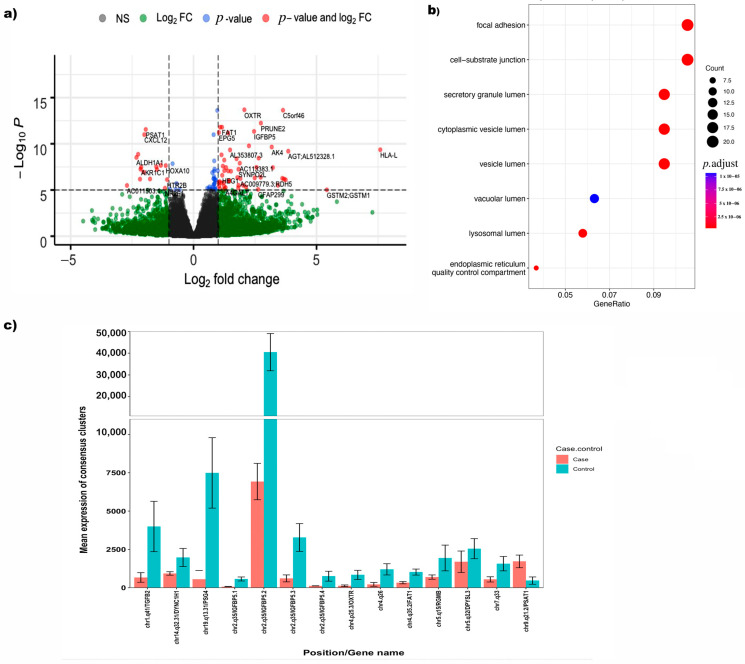
(**a**) Volcano plot for differential expression analysis of consensus clusters/promoters (from merged) between endometriosis controls and cases; (**b**) pathway analysis for 210 FDR significant consensus clusters; (**c**) top 14 differentially expressed consensus clusters in cases and controls obtained from the merged data.

**Figure 5 cells-12-01736-f005:**
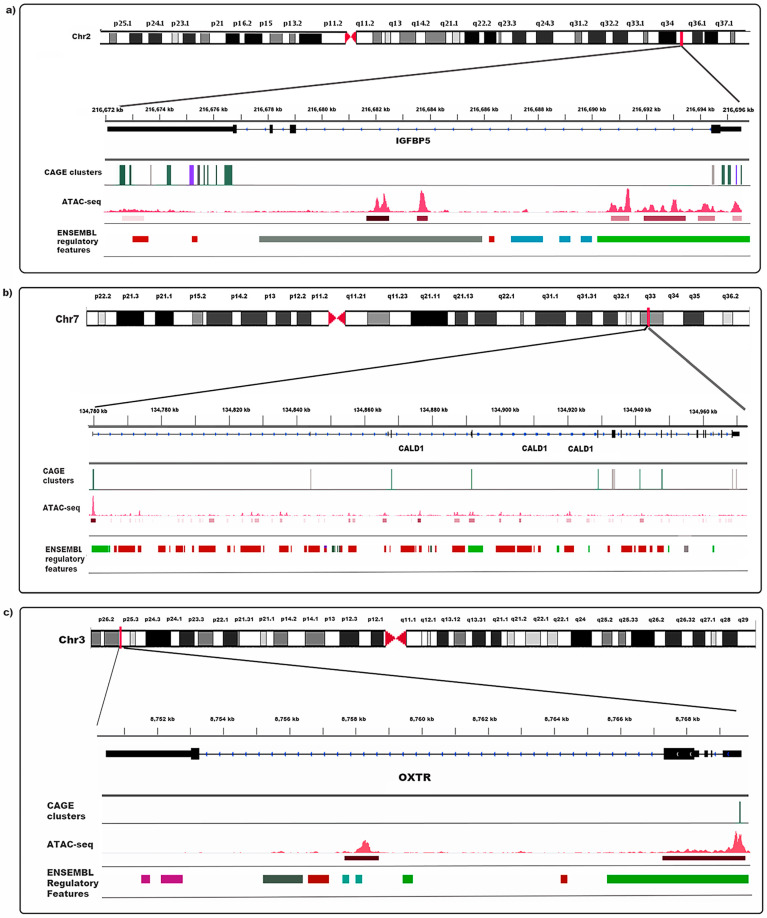
Maps of the chromosomal positions and gene context with consensus CAGE clusters annotated to the genes IGFBP5 (**a**), CALD1 (**b**) and OXTR (**c**), highlighting the overlap of cage clusters, open chromatin and Ensembl regulatory features. CAGE clusters are denoted by peaks on the CAGE clusters track. CAGE clusters in green were significantly differentially expressed following multiple testing correction (FDR *p* < 0.05) between endometriosis cases and controls, while clusters in purple were not annotated in FANTOM5. ATAC-seq signals are displayed as red peaks, and the corresponding narrow peak call is indicated by the red bars underneath the peaks. In the Ensembl regulatory features track, the following features are depicted as rectangular blocks with promoters (green), enhancers (brown), CTCF binding sites (blue) and open chromatin (pink). Where multiple features are clustered and would only be distinguished by enlarging the scale, these regions are denoted by dark green rectangular blocks.

**Figure 6 cells-12-01736-f006:**
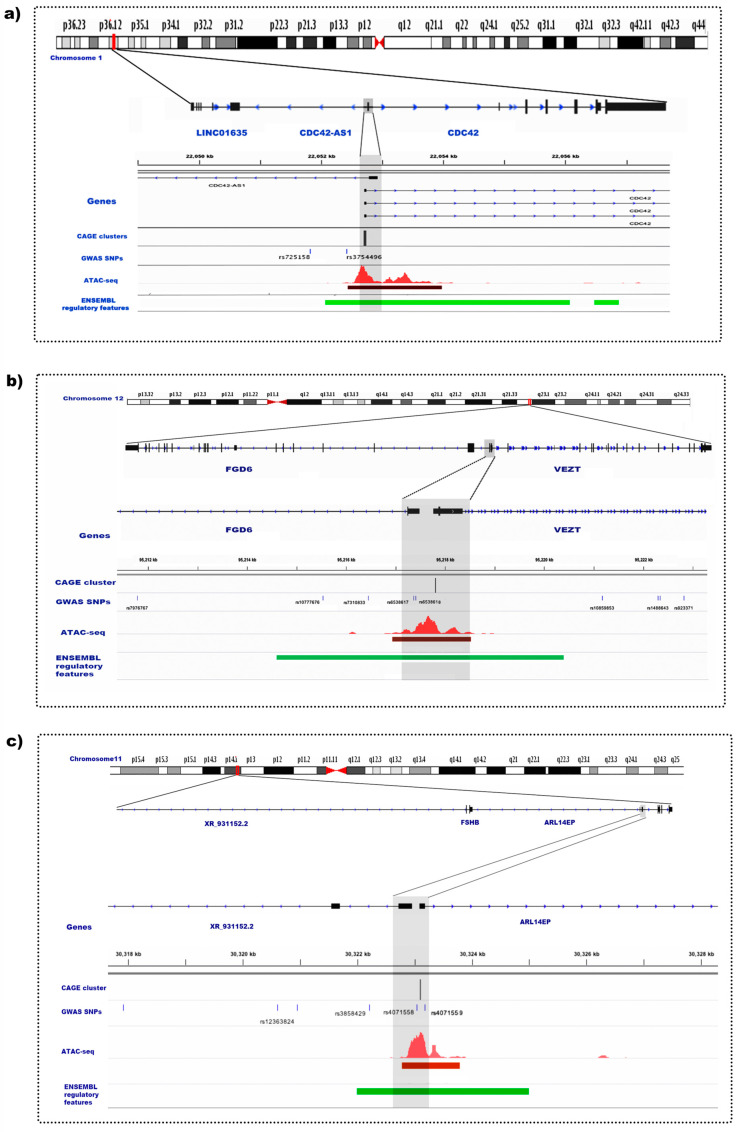
Maps of the chromosomal positions and gene context with consensus CAGE clusters and overlap with endometriosis risk SNPs in genomic regions for the CDC42 region (**a**), FGD6/VEZT region (**b**) and ARL14EP region (**c**). Location of SNPs significantly associated with endometriosis from GWAS studies are shown on the GWAS SNPs track with the corresponding SNP IDs. The promoter region of each gene is denoted by the grey-shaded area on the figures and demonstrates the overlap between CAGE clusters (peaks on the CAGE cluster track), open chromatin (red peaks on the ATAC-seq track with corresponding narrow peak call shown by the dark red bar underneath) and Ensembl promoter features shown in green on the Ensembl regulatory features track.

**Table 1 cells-12-01736-t001:** Information on clinical parameters for participants included in the CAGE analysis.

Sample ID	Case/Control	rAFS ^%^	ENZIAN ^&^	Age	BMI	Smoking ^#^	Infertility *
Sample 1	Case	I	0	43	25	0	0
Sample 2	Control	0	0	51	24.3	0	0
Sample 3	Control	0	0	41	24.2	0	1a
Sample 4	Case	I	A0B2C0	46	20.7	0	0
Sample 5	Case	I	0	36	22.9	1	1a
Sample 6	Case	II	2B	31	21.3	0	0
Sample 7	Control	0	0	25.7	28.7	n.d.	0
Sample 8	Control	0	0	21.7	17.2	0	0

^%^ rAFS score for endometriosis disease stage for cases were given according to the American Fertility Society scoring system (12). ^&^ ENZIAN scores provide a detailed description of the location and size or severity of the lesions and were given according to the Enzian classification (13). ^#^ Smoking status; 1 = Yes and n.d.= no data. * Infertility status; 1a = primary and 1b = secondary.

**Table 2 cells-12-01736-t002:** Number of CAGE tag counts obtained in each sample (women with endometriosis = cases; women without endometriosis = controls) from commercial and in-house protocols.

Sample ID	Case/Control	Commercial	In-House
Sample 1	Case	13,878,668	7,424,199
Sample 2	Control	11,976,955	6,373,486
Sample 3	Control	10,337,586	6,380,118
Sample 4	Case	13,202,981	6,579,239
Sample 5	Case	14,025,009	5,394,546
Sample 6	Case	14,100,380	7,617,671
Sample 7	Control	16,107,917	9,768,981
Sample 8	Control	20,367,598	7,825,180

**Table 3 cells-12-01736-t003:** Overlap of CAGE tag clusters with ATAC-seq data of endometrial stromal cells.

Sample_ID	Case/Control	Number of Tag Clusters	ATAC-Seq Peaks	No. of Tag Clusters Overlapping with ATAC-Seq
Sample 1	Case	8343	114583	6688
Sample 2	Control	8337	138093	6933
Sample 3	Control	8327	125615	6869
Sample 4	Case	8346	116957	6860
Sample 5	Case	8319	137215	7005
Sample 6	Case	8346	121518	6913
Sample 7	Control	8389	122149	6834
Sample 8	Control	8403	122214	6802

**Table 4 cells-12-01736-t004:** Number of CAGE tag clusters identified in each sample after merging commercial and in-house analysis data.

Sample ID	Case/Control	Tag Clusters
Sample 1	Case	8343
Sample 2	Control	8337
Sample 3	Control	8327
Sample 4	Case	8346
Sample 5	Case	8319
Sample 6	Case	8346
Sample 7	Control	8389
Sample 8	Control	8403

**Table 5 cells-12-01736-t005:** List of 53 consensus promoters differentially expressed in women without endometriosis and with endometriosis following Bonferroni correction for multiple testing (*p* < 7.02 × 10^−6^).

Chr	Start	End	Strand	TPM ^#^	Annotation	Genes	Log2FC	*p*-Value	FDR *
chr2	216694838	216694915	−	13.00	promoter	IGFBP5	3.05	3.00 × 10^−16^	2.13 × 10^−12^
chr3	8769610	8769628	−	26.97	promoter	OXTR	2.65	2.24 × 10^−15^	7.93 × 10^−12^
chr2	216695547	216695559	−	1476.3	promoter	IGFBP5	2.59	5.23 × 10^−14^	1.24 × 10^−10^
chr5	147906536	147906541	−	10.87	promoter	C5orf46	3.62	1.29 × 10^−12^	2.29 × 10^−9^
chr14	101964571	101964575	+	44.23	promoter	DYNC1H1	1.00	6.75 × 10^−12^	9.57 × 10^−9^
chr5	98773661	98773701	+	41.68	promoter	RGMB	1.35	1.23 × 10^−10^	1.45 × 10^−7^
chr19	43205633	43205648	−	108.38	promoter	PSG4	3.74	1.89 × 10^−10^	1.92 × 10^−7^
chr4	186726665	186726728	−	28.46	promoter	FAT1	1.58	2.54 × 10^−10^	2.25 × 10^−7^
chr2	216676450	216676709	−	80.13	exon	IGFBP5	2.47	3.49 × 10^−10^	2.75 × 10^−7^
chr4	114364911	114364951	−	27.86	unknown		2.37	8.15 × 10^−10^	5.78 × 10^−7^
chr2	216674321	216674452	−	20.13	exon	IGFBP5	2.76	1.67 × 10^−9^	1.08 × 10^−6^
chr1	218345334	218345346	+	76.83	promoter	TGFB2	2.44	3.02 × 10^−9^	1.78 × 10^−6^
chr7	134711476	134711515	+	34.95	unknown		1.45	3.99 × 10^−9^	2.17 × 10^−6^
chr9	78297122	78297161	+	36.46	promoter	PSAT1	−1.94	6.68 × 10^−9^	3.38 × 10^−6^
chr3	188212669	188212713	+	37.34	promoter	LPP	1.858	1.19 × 10^−8^	5.62 × 10^−6^
chr7	134646835	134646861	+	29.33	promoter	BPGM	0.95	1.91 × 10^−8^	8.47 × 10^−6^
chr2	216695059	216695140	−	8.79	promoter	IGFBP5	2.65	3.88 × 10^−8^	1.62 × 10^−5^
chr4	94451901	94451973	+	109.58	promoter	PDLIM5	0.83	5.87 × 10^−8^	2.31 × 10^−5^
chr6	26021572	26021654	+	23.35	promoter		−1.43	6.45 × 10^−8^	2.40 × 10^−5^
chr5	40679914	40679919	+	5.58	promoter	PTGER4	−1.22	7.21 × 10^−8^	2.55 × 10^−5^
chr16	3065631	3065641	+	6.75	promoter	IL32	2.33	9.46 × 10^−8^	3.19 × 10^−5^
chr9	72953039	72953073	−	30.88	promoter	ALDH1A1	−2.26	2.21 × 10^−7^	7.13 × 10^−5^
chr6	131949556	131949569	−	4.73	exon	CCN2	1.85	2.71 × 10^−7^	8.33 × 10^−5^
chr17	80260832	80260882	+	19.59	promoter	AC124319.1	1.02	2.94 × 10^−7^	8.69 × 10^−5^
chr12	56315875	56316039	−	86.96	promoter	AC073896.1; CNPY2	-0.44	3.73 × 10^−7^	0.00010
chr18	45967267	45967330	−	21.44	promoter	EPG5	0.838	4.59 × 10^−7^	0.00012
chr2	216675681	216675683	−	2.52	exon	IGFBP5	2.85	4.70 × 10^−7^	0.00012
chr15	63042680	63042756	+	659.91	promoter	TPM1	0.86	4.96 × 10^−7^	0.00012
chr2	216676128	216676131	−	3.19	exon	IGFBP5	3.59	5.28 × 10^−7^	0.00012
chr7	134867737	134867770	+	7.57	exon	CALD1	1.49	6.20 × 10^−7^	0.00014
chr5	178204530	178204537	+	114.22	promoter	HNRNPAB	−0.75	7.73 × 10^−7^	0.00017
chr9	38392671	38392799	+	44.95	promoter	ALDH1B1	1.954	9.92 × 10^−7^	0.00021
chr9	5510498	5510556	+	16.50	promoter	PDCD1LG2	1.149	1.25 × 10^−6^	0.00026
chr3	156674590	156674634	+	27.80	promoter	TIPARP	0.93	1.29 × 10^−6^	0.00026
chr8	23404118	23404156	−	732.82	promoter	LOXL2; ENTPD4	0.823	1.43 × 10^−6^	0.00028
chr12	29783910	29783922	−	17.55	promoter	TMTC1	−1.76	1.55 × 10^−6^	0.00029
chr8	41309471	41309474	−	3.24	promoter	SFRP1	−2.64	1.53 × 10^−6^	0.00029
chr5	139293674	139293754	+	75.81	promoter	MATR3	-0.53	1.61 × 10^−6^	0.00029
chr5	84384380	84384483	−	19.82	promoter	EDIL3	1.62	1.89 × 10^−6^	0.00034
chr20	50190829	50190835	+	51.41	promoter	CEBPB	−0.855	2.31 × 10^−6^	0.00040
chr11	62546749	62546845	−	101.41	promoter	AHNAK	1.031	2.43 × 10^−6^	0.00041
chr7	134928752	134928863	+	21.03	exon	CALD1	0.99	2.80 × 10^−6^	0.00047
chr11	117204261	117204391	+	42.70	exon	TAGLN	0.93	2.98 × 10^−6^	0.00047
chr5	141969105	141969137	+	6.875	promoter	RNF14	1.66	2.95 × 10^−6^	0.00047
chr10	32957884	32957980	−	58.70	promoter	ITGB1	1.192	3.16 × 10^−6^	0.00049
chr20	63696646	63696657	+	16.99	promoter	RTEL1-TNFRSF6B; TNFRSF6B	1.405	3.96 × 10^−6^	0.00060
chr1	109687817	109687847	+	4.17	promoter	GSTM2; GSTM1	5.476	4.47 × 10^−6^	0.00067
chr16	71358723	71358731	+	93.20	promoter	CALB2	2.51	4.96 × 10^−6^	0.00073
chr2	30231709	30231716	+	7.77	promoter	LBH	1.24	5.44 × 10^−6^	0.00078
chr2	216695357	216695370	−	52.35	promoter	IGFBP5	2.43	5.64 × 10^−6^	0.00079
chr2	216372053	216372078	−	6.62	promoter	MARCHF4	1.08	5.83 × 10^−6^	0.00080
chr1	78004920	78004954	+	30.40	promoter	DNAJB4; GIPC2	1.31	5.90 × 10^−6^	0.00080
chr9	116153791	116153813	+	44.29214	promoter	PAPPA	1.59	6.49 × 10^−6^	0.00086

^#^ TPM—tags per million; * FDR—false discovery rate.

**Table 6 cells-12-01736-t006:** Overlap of endometriosis GWAS SNPs within 2 kb genomic window from the identified CAGE tag clusters.

Chr	BP	SNP	*p*-Value	Allele	Gene Region
chr1	22051787	rs725158	4.88 × 10^−16^	a	CDC42
chr1	22052387	rs3754496	4.99 × 10^−16^	a	CDC42
chr2	215433073	rs1250244	8.93 × 10^−8^	c	FN1
chr7	137345599	rs161335	4.67 × 10^−6^	t	next to PTN
chr11	30322210	rs3858429	5.45 × 10^−8^	t	XR_931152.2
chr11	30323044	rs4071558	5.62 × 10^−8^	t	near ARL14EP
chr11	30323178	rs4071559	5.60 × 10^−8^	t	ARL14EP
chr12	95216444	rs7310833	8.07 × 10^−9^	a	FGD6
chr12	95217365	rs6538617	3.60 × 10^−6^	t	FGD6
chr12	95217409	rs6538618	6.68 × 10^−9^	t	FGD6
chr17	44898335	rs35653192	9.15 × 10^−6^	a	EFTUD2

## Data Availability

The data underlying this article will be shared on reasonable request to the corresponding author.

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
