# Peer review of "Global Analysis of Transcription Start Sites and Enhancers in Endometrial Stromal Cells and Differences Associated with Endometriosis"

_cells, 2023, doi:10.3390/cells12131736_

Round 1
Reviewer 2 Report
Dear authors,
I read with great interest the manuscript, which falls within the aim of this Journal. In my honest opinion, the topic is interesting enough to attract the readers’ attention. Nevertheless, authors should clarify some points and improve the discussion, as suggested below. Authors should consider the following recommendations:
In my opinion you have to improve the paper refering in the text to the fertility sparing treatments (FST) to the updated literature especially in pts that have to preserve their fetility for oncological reason and its implication.
Also about endometriosis I suggest you read and cite these interesting paper suggested below
Biomolecular and Genetic Prognostic Factors That Can Facilitate Fertility-Sparing Treatment (FST) Decision Making in Early Stage Endometrial Cancer (ES-EC): A Systematic Review
Genome-Wide DNA Methylation Analysis Predicts an Epigenetic Switch for GATA Factor Expression in Endometriosis
Fertility preservation in female cancer sufferers: (only) a moral obligation?
Minor editing of English language required
Reviewer 3 Report
Its a well planned, multi-platform integrative and systematically conducted research. The study provides clear evidence emphasizing importance of mapping cell-type specific transcriptional events for endometriosis. One advice is that if the authors provide a graphical abstract (summarizing the methodology), it will help the readers to understand the methodology adopted and the prominent findings.
